behaviour, ecology, evolution

*Gorilla*, social structure, hierarchical, modularity, community, multi-level

**Author for correspondence:**
Robin E. Morrison
e-mail: robinemilymorrison@gmail.com

# Hierarchical social modularity in gorillas

Robin E. Morrison[1], Milou Groenenberg[2], Thomas Breuer[2,3], Marie L. Manguette[2,4] and Peter D. Walsh[5]

[1]Department of Archaeology, University of Cambridge, Downing Street, Cambridge CB2 3DZ, UK
[2]Mbeli Bai Study, Wildlife Conservation Society - Congo Program, B.P. 14537 Brazzaville, Republic of Congo
[3]World Wide Fund for Nature, Reinhardtstrasse 18, 10117 Berlin, Germany
[4]Max Planck Institute for Evolutionary Anthropology, Deutscher Platz 6, 04103 Leipzig, Germany
[5]Apes Incorporated, 5301 Westbard Circle, Bethesda, MD 20816, USA

REM, 0000-0001-9161-4734; TB, 0000-0002-8387-5712; MLM, 0000-0003-2388-3267

Modern human societies show hierarchical social modularity (HSM) in which lower-order social units like nuclear families are nested inside increasingly larger units. It has been argued that this HSM evolved independently and after the chimpanzee–human split due to greater recognition of, and bonding between, dispersed kin. We used network modularity analysis and hierarchical clustering to quantify community structure within two western lowland gorilla populations. In both communities, we detected two hierarchically nested tiers of social structure which have not been previously quantified. Both tiers map closely to human social tiers. Genetic data from one population suggested that, as in humans, social unit membership was kin structured. The sizes of gorilla social units also showed the kind of consistent scaling ratio between social tiers observed in humans, baboons, toothed whales, and elephants. These results indicate that the hierarchical social organization observed in humans may have evolved far earlier than previously asserted and may not be a product of the social brain evolution unique to the hominin lineage.

## 1. Background

How did human society transition from small, autonomous groups, to multi-tiered, hierarchically nested structures in which networks of association and cooperation coalesce into successively higher-level units? And when did this happen? According to the dominant narrative, the transition to a complex, multi-tiered society in humans was part of a broader trend in mammalian evolution in which brain size increase is associated with a suite of social cognition capacities referred to as the Social Brain [1]. This hypothesis is supported by the presence of highly developed neo-cortices in the mammalian taxa in which hierarchical social modularity (HSM) is best documented (primates, elephantidae, odontocetes) [2,3]. In fact, all of these taxa show a similar scaling pattern in which the size of social groups at each social tier is the same fixed multiple of the size of groups in the next lower tier, implying some common mechanism is at play [4].

An extreme version of the social brain narrative is that human HSM is a product of hominin brain evolution, distinct from HSM observed in other primates [5], kick-started when early hominins living in multi-male, multi-female societies evolved a heightened capacity to recognize dispersed kin [6]. The social brain enhancements that evolved in the context of collaborating with large coalitions of dispersed kin then facilitated the development of HSM when the transition to single male social groups brought more structure to kin interactions. In this narrative, the extension to even larger networks of reciprocity among non-kin is due to further expansion of social brain capacity [7]. A major foundation of this argument is that of all great apes, only humans have been documented to show HSM [5]. But is the absence of evidence, evidence of absence? Although chimpanzees appear not to show HSM, there has been no rigorous attempt to study HSM in the next sister

taxon, gorillas. And there are good reasons to suspect that western gorilla (*Gorilla gorilla*) may exhibit HSM. Like most traditional human societies [8], western gorillas predominantly live in reproductive groups with only one adult male (silverback) and one or more females with dependent offspring [9]. Upon reaching sexual maturity, both sexes disperse from their natal groups [10]. Females transfer into another social group, however, male gorillas may spend many years as solitary males before they are able to attract their own females and form a stable social group [9]. Unlike chimpanzees, strong territoriality does not prevent higher-order associations between social units. Rather, much like humans, western gorilla reproductive groups occupy overlapping home ranges [11,12] and often aggregate at resource hotspots [13]. There are also anecdotal reports of affiliative interactions between these groups [12,14] and genetic evidence that individuals may regularly move between groups [15] and that silverback males may choose to live in close proximity to related silverbacks [16].

HSM has not been previously studied in western gorillas in good part because their home ranges span large swaths of thick tropical forest, making observations of inter-group social interaction difficult. To circumvent this problem, we analyse observational data from two mineral-rich forest clearings in the Republic of Congo. The superabundant resources in such clearings draw gorillas from considerable distances, creating hubs for social interaction [17,18]. Hierarchical clustering and network modularity analyses are used to evaluate whether gorilla visit patterns indicate a modular social organization, with higher-level social units formed of multiple gorilla groups and solitary males. We then evaluate whether the sizes of gorilla social units show hierarchical scaling [4] and use genetic data to test whether these social units have kin structure similar to humans.

## 2. Methods

### (a) Generating networks

Two long-term datasets of western lowland gorilla visits to forest clearings (known locally as bais) in the Republic of Congo were used in the analysis. Gorillas are attracted to these forest clearings by the mineral- and protein-rich vegetation [17,18], on which they usually feed for many hours at a time, allowing individual gorillas to be identified and studied from research platforms located on the edge of these clearings [9,19,20]. The Lokoué published dataset [20], covers a period of 409 days from April 2001 to September 2002 and includes visit data on 205 individuals forming 48 gorilla units (27 groups and 21 solitary males). The Mbeli dataset [9,10,19] is formed of data collected from January 2010 to December 2015 when the bai was monitored almost daily (2191 days) and includes visit data on 271 individuals, forming 44 gorilla units (19 groups, 18 solitary males, and 7 solitary males that formed groups during the study period). The Mbeli dataset was split into three separate 2-year datasets (A:2010−2011, B:2012−2013, and C:2014−2015) of 730, 731, and 730 days respectively. Individuals that visited fewer than eight times within a 2-year dataset were removed from the analysis. Values of association between all pairs of groups or solitaries were calculated using the forest clearing visit data whereby any visit by a pair of groups or solitaries on the same day (covisit) was counted as an association. Association values were calculated via two complementary methods. Firstly, the Asnipe R package [21] was used to generate simple ratio (SR)

association indices (the most widely used form of association index in social network analysis) and generate null models to compare these to, through data stream permutations. Secondly, the binomial probability (BP) association index was calculated, and null models generated as specified in the electronic supplementary material. The BP index calculates the probability that two social units would covisit the observed number of times given the average visit rate by each unit during a given study interval. The BP index enabled us to correct for fluctuations in the rate at which all gorillas visited each clearing and, therefore, the probability of a random covisit. Use of the BP index also avoided one major problem with the standard SR association index, the potential for units with small numbers of visits to be spuriously assigned very high association values. Networks were produced using both association indices for all datasets and null models using the 'igraph' R package [22]. Agreement in pairwise association values across consecutive time periods was investigated using a mantel test in the 'ape' R package with 1000 permutations.

### (b) Detection of hierarchical modular structure

Modular structure in the Lokoué and Mbeli C networks was initially investigated with a hierarchical clustering approach [23] using the 'cluster' R package and 'average' (UPGMA) method. This produced hierarchical dendrograms, where groups and solitaries that associated most strongly with each other were joined on the dendrogram over the shortest distances on the $y$-axis. This analysis was done using the BP association values (electronic supplementary material) and the distances at which every join (bifurcation) occurred in the dendrogram were extracted. Association values were transformed ($x^{2/3}$). This enabled the rate of cumulative bifurcations (total number of joins in the dendrogram with distance) in null models to fit a linear relationship, such that any change in gradient within the observed data would represent a transition from one level of social structure to the next. Changes in gradient (knots) were identified by Wilcoxon Two Sample Test and $R^2$-values from linear regression were used to compare the fit of real and random networks.

Higher tier social units using both the SR and BP association indices were identified using the Louvain multi-level modularity optimization algorithm [24], which searched for modules of gorilla groups or solitaries that were more strongly associated with one another than the wider gorilla population. Modularity from each dataset was compared with that from 1000 null models to generate $p$-values. Consistency between methods was investigated using a binomial linear model to predict co-membership of the same higher tier module (detected by modularity analysis) from co-membership of the same lower tier cluster (detected by hierarchical clustering).

As the previous modularity algorithm was designed to detect a single optimal level of modularity, and therefore a single level of social structure, an alternative method was required to search for multiple levels of social structure within a population. The cluster_resolution 'igraph' algorithm [25] was therefore used to search for multiple peaks in modularity at varying module size (indicating multiple levels of social structure), as a complementary approach to confirm the social tiers detected by hierarchical clustering. This was done by varying the algorithm's resolution parameter between 0 and 2 by increments of 0.01, altering how strong links within a module of groups and solitaries must be to assign a discrete module and, therefore, the number of modules in a given population. $p$-values were then calculated as the proportion of null models at the same resolution value with modularity values equal to or greater than that of the data. This analysis was run on SR associations only, as $p$-values produced using the BP method were too low to show adequate variation with a feasible number of null models.

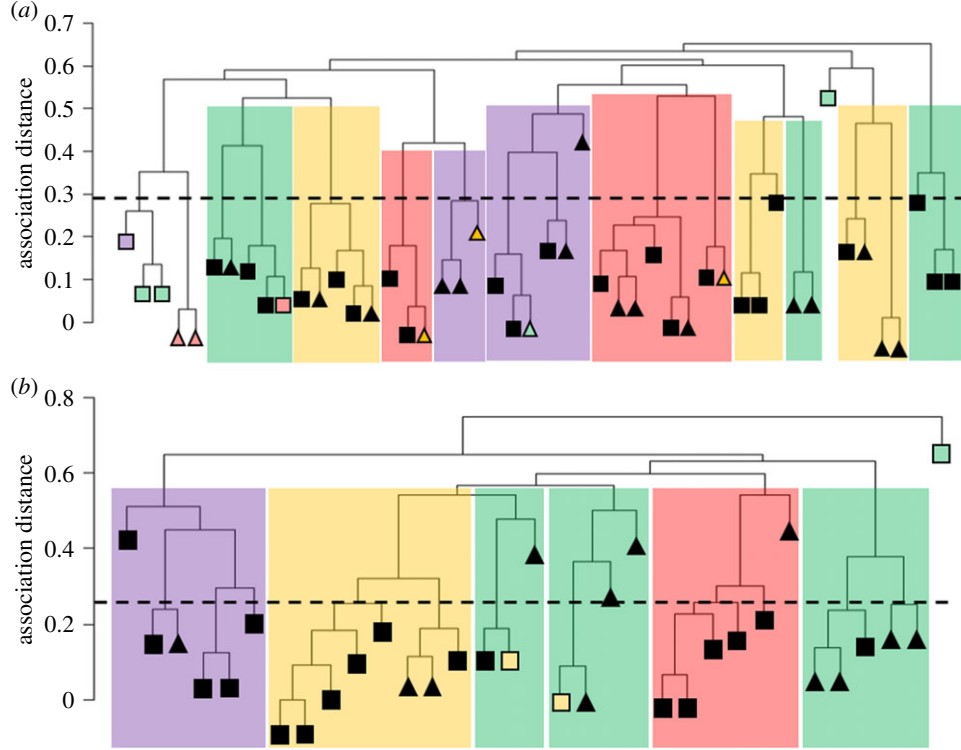

**Figure 1.** Multi-level structure of the Lokoué (a) and Mbeli (b) populations produced by hierarchical clustering using the BP association index, show preferential associations between gorilla groups and solitary males. Height of significant knot (upper limit of first social tier) indicated by dashed line, such that groups or solitaries joined before the dashed line are within the same first-order social unit. Social units detected by modularity analysis indicated by background shading, such that those with the same background shading are within the same second tier social unit. Squares indicate groups, triangles indicate solitary males. Disagreements between groupings by hierarchical clustering and modularity analysis indicated with colour of triangles or squares. For Mbeli Bai, the C dataset was used. (Online version in colour.)

**Table 1.** Modularity values for all four networks by association index. *p*-values (in brackets) calculated by comparison with 1000 networks built from permutations of the original data demonstrate that real networks show higher modularity than expected by chance.

| | simple ratio | binomial probability |
|---|---|---|
| Lokoué | 0.191 (<0.001) | 0.040 (0.001) |
| Mbeli A (period 2010–2011) | 0.104 (0.069) | 0.055 (0.003) |
| Mbeli B (period 2012–2013) | 0.091 (0.03) | 0.047 (0.077) |
| Mbeli C (period 2014–2015) | 0.082 (0.009) | 0.052 (0.025) |

### (c) Scaling

Scaling of social unit size was investigated using results from the Lokoué and Mbeli C (the Mbeli dataset of largest population size (electronic supplementary material, table S1)) datasets, as well as published data from Maya-Nord [26]. The log of social unit size at each social level was taken. Linear models were run to predict log social unit size by social level, producing *R*-squared and *p*-values, for social unit sizes detected by each method (Method A: hierarchical clustering approach and Method B: modularity resolution varying approach). This was done for all three populations separately and then combined while controlling for the specific population.

### (d) Kinship

Published binary kinship data [20] of silverback male pairwise relatedness from the Lokoué population (*n* = 20) were used to predict co-membership of the same social unit using binomial logistic regression. Binary genetic data were based on microsatellite markers from DNA extracted from faecal samples, with 1 indicating an estimated relatedness of greater than or equal to 0.2 and 0 indicating an estimated relatedness of less than 0.2. This cut-off should assign all pairs that are half-siblings (relatedness = 0.25) or more closely related, a value of 1, with some room for error in estimate precision. The social units used in these analyses were the higher tier modules detected by multi-level modularity analysis using the SR index and BP index.

## 3. Results

Association indices were generated using visits of gorilla groups and solitary males to a clearing on the same day, as a metric of social association, to estimate the probability of both visual contact and long-distance auditory signalling (e.g. chest beating) between them [27]. We applied clustering analyses [23] to the BP association index based on the probability that pairs of groups or solitaries would appear at a clearing on the same day. The resulting dendrograms (figure 1) showed a pattern of preferential association between small clusters of groups and solitaries. Analyses of the rate at which bifurcations accumulated with association distance (*d*) when moving from tip to base of each dendrogram placed significant knots at *d* = 0.29 (*w* = 722, *p* = 0.0147) for Lokoué and *d* = 0.26 (*w* = 498.5, *p* = 0.0285) for Mbeli (electronic supplementary material, figure S1), indicating a transition from one social tier to the next. The number of groups or solitaries involved in the first tier of associations (below the social tier transition) averaged 2.29 for Lokoué

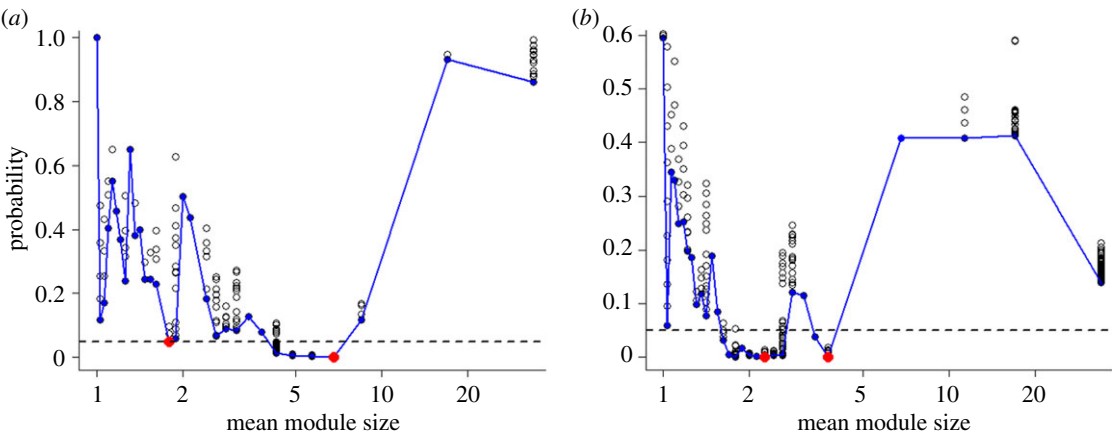

**Figure 2.** p-values of modularity scores for a given size of module for (a) Lokoué and (b) Mbeli (using dataset (C)), produced by varying the modularity resolution parameter demonstrate two p-value troughs, suggesting two separate levels of modularity (and hence social structure) in the associations between gorilla groups and solitary males. Most significant value in both troughs of probability indicated in red. (Online version in colour.)

and 1.94 for Mbeli and included a weighted mean of 11.4 individual gorillas.

Modularity analyses were then used to further investigate these associations. The strongest modularity signal detected using the SR index was for a previously unreported second tier of association involving a weighted average of 39.3 individual gorillas made up of 8.1 independent groups or solitaries (8 at Lokoué, 8.25 at Mbeli (using the Mbeli C dataset)). Statistical support for modularity at this level was very strong for Lokoué and two of the Mbeli sampling intervals and weaker for a third Mbeli sampling interval (table 1). After correcting for seasonal variation in visitation rates, the association index based on the BP of same day visits still produced strong statistical support, similar to that from the classic SR index, suggesting that environmental variables within the bai were unlikely to be driving the observed pattern. At Mbeli, pairwise associations between group and solitary gorillas using the SR index were highly consistent between consecutive time periods (Mantel test: 2010–2011 with 2012–2013: $Z = 0.355$ $p = 0.002$, 2012–2013 with 2014–2015: $Z = 0.663$ $p = 0.001$), and even non-consecutive time periods (Mantel test: 2010–2011 with 2014–2015: $Z = 0.341$ $p = 0.004$), suggesting long-term stability in affiliative relationships rather than short-term competitive interactions, for example, in the acquisition of reproductive females [28]. For both populations, membership of pairs of groups or solitaries in the same first tier associations detected by clustering, strongly predicted their presence in the same second tier associations detected by modularity analysis (Lokoué $z = 7.144$ $Pr(>|z|) < 0.0001$, Mbeli $z = 5.245$ $Pr(>|z|) < 0.0001$), demonstrating consistency between the two approaches and that the structure detected was hierarchically inclusive.

The initial modularity algorithm used was designed to detect a single optimal level of modularity and biased upwards in the size of modules it detects. Therefore, to search for multiple peaks in modularity we manually varied the algorithm's resolution parameter, which defines how relatively strong links between nodes (in this case, gorilla groups and solitary males) must be to assign a discrete module and, therefore, the number of modules in a given population [25]. For both study populations, using the SR index, this revealed a second peak in modularity (trough in random probability) at an average of 2.03 groups or solitaries per association, or 13.1 individuals (figure 2 and electronic

supplementary material, figure S2); a first tier association size very similar to that suggested by clustering analysis using the BP index.

The tiers of western gorilla social hierarchy detected by clustering and modularity analyses correspond closely to those of other HSM taxa. The first tier associations we record, involving a mean of 13.1 gorillas, map closely to tier g3 (dispersed extended family group) in Binford's classification of traditional human societies [29,30], where g1 and g2 are, respectively, individuals and nuclear family groups. They also resemble baboon 'clans', gelada 'teams', elephant 'bond groups', and dolphin 'first-order alliances' [4]. The new, second tier of association we detect, involving a mean of 39.3 gorillas maps to Binford's g4 (aggregated group), baboon and gelada 'bands', elephant 'clans', and dolphin 'second-order alliances'. The potential for Binford's tier g5, periodic aggregations at resource hotspots, is demonstrated in the tendency for many gorilla groups to converge on places like Mbeli during super-annual 'mast' fruiting events [13]. However, our observation days were too few to provide adequate statistical power for detecting this tier. Community closure consistent with Binford's tier g6 (population) is indicated by asymptotic new group accumulation curves at Lokoué and Mbeli (electronic supplementary material, figure S3). Gorillas also exhibit an association grade observed in humans and referenced by animal ecologists but omitted by Binford's classification, preferential affiliation within mother–offspring units [31,32].

When group size in each social tier is approximated by an exponential function, the goodness of fit is extremely high for both Lokoué and Mbeli ($R^2 = 0.996$ and $0.994$; figure 3), indicating that group size at each social tier increases by a consistent multiplier relative to group size at the next lower tier. The estimated scaling exponents for the two sites (2.78 and 2.73) were similar both to each other and that estimated from a nearby site, Maya Nord (3.07) where data on social group and population size but not rates of association are published [26]. A slightly lower scaling exponent than for other HSM taxa [4] is consistent with a lower demographic rate that produces fewer potential kin associates.

Published data on binary genetic relatedness of pairs of Lokoué silverbacks (group leaders or solitary males) predicted their joint membership in the second-order associations (modules) detected by modularity analysis

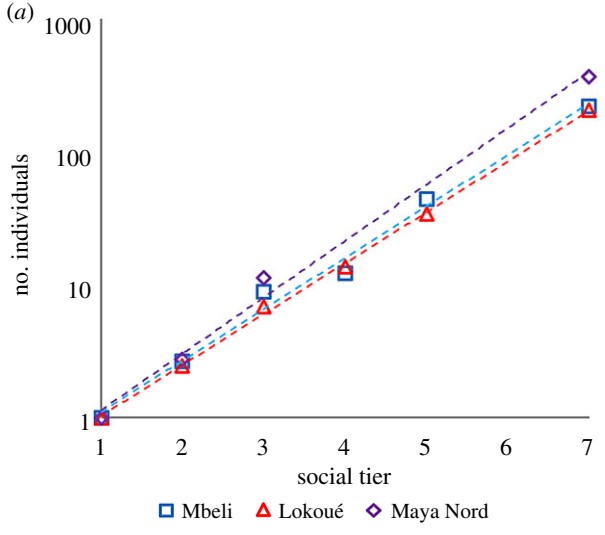

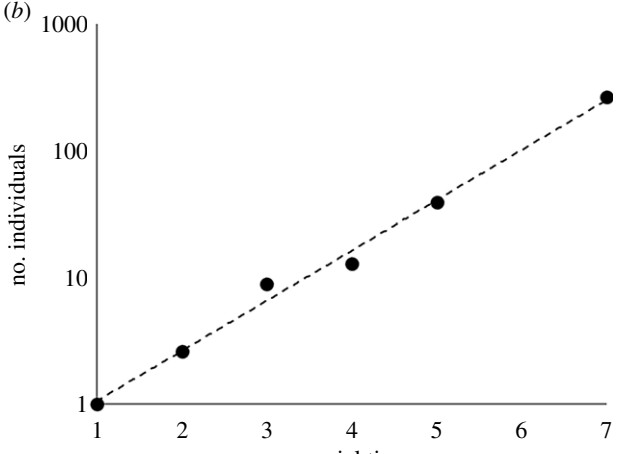

**Figure 3.** Social unit sizes across three gorilla populations follow a consistent scaling relationship close to that observed in other mammalian species showing HSM. Social tier values 1–7 represent g1 (individuals), mother–offspring units, g2 (family units), g3 (dispersed extended family group), g4 (aggregated group), g5 (sub-population), and g6 (overall population). (*a*) For three separate populations Mbeli (blue), Lokoué (red), Maya Nord (purple) with their fitted exponentials shown by dashed lines. (*b*) For the mean of social unit sizes at each level from all populations, with dashed line indicating the fitted exponential (scaling ratio = 2.70, $R^2 = 0.991$, $p = 6.37 \times 10^{-10}$). (Online version in colour.)

(mean size = 8.1 groups or solitaires, 39.3 individuals), with silverback males that were half-siblings or more closely related, more likely to be within the same module. Joint membership of modules calculated using the BP index ($z = 2.0$, $\Pr(>|z|)$ 0.045) was better predicted by relatedness than modules from the SR index ($z = 1.8$, $\Pr(>|z|)$ 0.072). This is consistent with kinship influencing the underlying pattern of associations with some additional variation introduced from environmental variables. However, related silverbacks represented only 14.6% and 12.8% of total pairs within the same module for which relatedness was known, for BP and SR modules, respectively, demonstrating that a considerable proportion of the associations detected were occurring between males less closely related than half-siblings. Due to the dispersal of females between groups multiple times throughout their lives, it is possible for males that are not closely related to grow up within the same natal group. This may then lead to associations as adults. Alternatively, associations between unrelated males could be due to their

presence in distinct but closely associated natal groups or develop during periods post-dispersal when young males have been known to form all-male bachelor groups [33]. The first order of associations (mean size = 2.03 groups or solitaires, 13.1 individuals) could not be investigated genetically due to the low number of silverbacks within the same first-order association for which genetic data were available.

# 4. Discussion

Our results suggest a social structure in western gorillas with striking parallels to human society, from the kin bias of social modules, to the hierarchical scaling in their size, providing a potential link between human social structure and the modular societies observed in many other primate species [5]. Given the likely presence of HSM in both humans and gorillas, and its relatively rare occurrence across mammalian species, our results suggest it is more parsimonious to assume that HSM evolved in a common ancestor of gorillas and humans and was lost in chimpanzees, rather than evolving independently in both lineages (figure 4). This may also be the case for the predominance of single male reproductive groups in humans which may have been inherited from the common ancestor of all apes, rather than being replaced by a territorial mm-mf structure in the most recent common ancestor of chimpanzees and humans and then regained in the hominin lineage (electronic supplementary material, figure S4). Given that gorillas and humans also share characters such as a matrix of evenly spaced, overlapping home ranges and long-tailed distributions of social contact at resource hotspots [13], they may have considerable advantages as a model system for human social evolution, relative to the more frequently relied upon chimpanzee (*Pan troglodytes*) [6,34]. However, primate social systems are so plastic within and variable between species that it seems imprudent to lean too heavily on the states of extant taxa when drawing conclusions about distantly related early hominins. Rather, our key point is that if we explicitly define 'complexity' as nested hierarchical structure, then the social brain enhancements of the hominin neocortical explosion do not appear necessary to explain human social complexity. What remains unclear is the role of the social brain in the broader pattern of mammalian HSM. This lack of clarity stems from the fact that although characters like large brain size and extended parental dependence are often attributed to the intricate social lives of taxa that exhibit HSM [7,35] they can also be explained in terms of the profound effect that large body size has on foraging ecology [36].

HSM taxa use the fasting capability granted by large body size to exploit rare, transient, and unpredictable resource hotspots [36]. Western gorillas, baboons, and forest elephants move kilometres each day to feed at diverse tree species that fruit on sporadic 'mast' schedules [13,37–39]. Odontocetes travel even further to hunt fish schools that form and dissolve in equally idiosyncratic ways [40]. Efficiently implementing this strategy over a large home range requires copious spatial memory for recording resource location. Consequently, gorillas, elephants, and odontocetes have spatial memory centres that rival or exceed humans' [41]. This strategy also relies heavily on associative learning. Thus, the delayed natal dispersal that typifies HSM taxa is not just for mastering specific skills [42,43]. Rather,

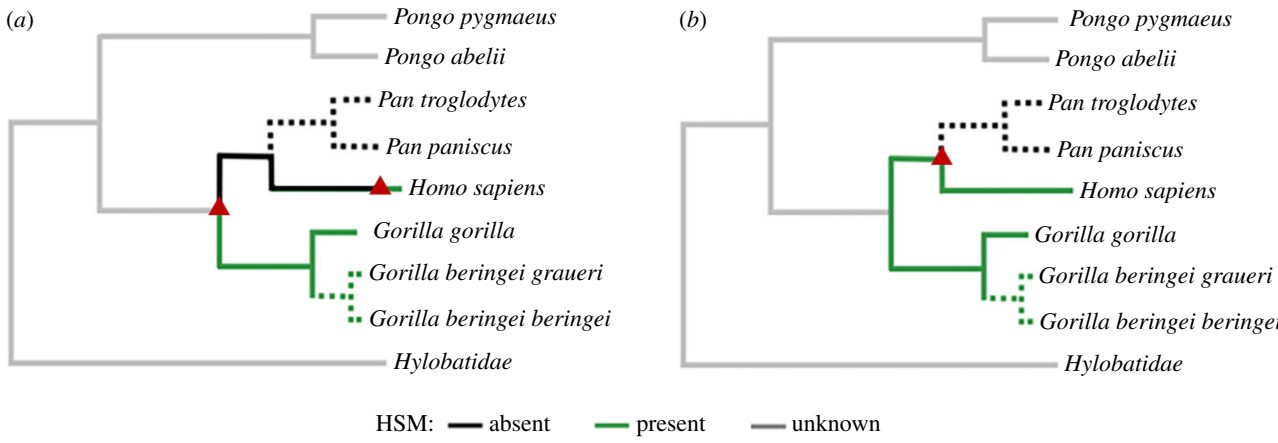

**Figure 4.** The presence of HSM in ape species plotted on a phylogeny of apes [11]. Suspected but unconfirmed traits indicated by dashed lines, transitions indicated by triangles. (*a*) Transitions required under the assumption that HSM evolved late in the hominin lineage. (*b*) Transitions required under our proposed, more parsimonious model, of social evolution. (Online version in colour.)

specializing on patchily distributed resources with idiosyncratic dynamics may inherently require long training periods. Delayed natal dispersal in HSM taxa generates overlapping cohorts of offspring and a ready pool of possible cooperators, with the potential for cooperative foraging to dramatically increase resource localization rate [44,45]. This could be coordinated through the long distance, individually recognizable contact calls common to HSM taxa [46,47]. Long-term associations between siblings in natal groups could provide the kind of repeat interaction critical to reputation building and the stabilization of reciprocity networks [48]. If dispersing offspring establish home ranges adjacent to their parents or siblings [16,49,50], rates of exposure would correlate with degrees of relatedness and the strong social bonds formed within natal groups. This could enable considerable reciprocal and kin-based benefits to cooperation between neighbouring groups for foraging, and a potential driver for the evolution of the complex HSM present in these species.

Our results demonstrate extensive, previously overlooked similarities between human and gorilla social structure, suggesting that the social brain enhancements observed within the hominin lineage were not necessary to enable HSM. When contextualized with common trends across other mammalian HSM taxa, the remarkable consistency in life-history and foraging behaviour is suggestive of a possible alternative mechanism as a driver of social complexity; that of collaborative foraging. Under this hypothesis, selection for optimizing foraging strategies when feeding on patchily distributed and unpredictable resources could drive the maintenance of affiliation and communication between dispersed kin, and ultimately the complex HSM observed in these lineages. While further research is clearly required to investigate this hypothesis, the presence of a kin-based,

multi-tiered social structure in gorillas suggests that fundamental elements of human social complexity may have far deeper evolutionary roots than previously assumed, and that understanding the mechanistic details of how they emerged will require peering more deeply into our evolutionary past.

Data accessibility. Our paper uses pre-existing datasets from two gorilla research sites. Data from the Lokoué site is published in the doctoral thesis of Florence Levrero, available online and cited in text. Data from the Mbeli site has been used in previous publications but the raw data itself is not available due to the strong data regulations imposed by the Wildlife Conservation Society to limit poaching. Therefore to enable reproducibility, the association matrices constructed from the raw gorilla visit data, on which the analysis in this manuscript relies have been made available through the electronic supplementary material.

Authors' contributions. P.D.W. conceived the project; R.E.M. and P.D.W. developed the methodology; M.G., T.B., and M.L.M. managed data collection and designed data collection protocols; R.E.M. analysed the data with support from P.D.W.; R.E.M., and P.D.W. wrote the paper with support and feedback from all authors.

Competing interests. We declare we have no competing interests.

Funding. This work was supported by grants from the International Society for Human Ethology, the Nacey Maggioncalda Foundation, and the University of Cambridge.

Acknowledgements. We thank the Ministère de l'Economie Forestière of the Government of Congo and the Nouabalé-Ndoki Foundation for their permission to carry out this research as part of the Mbeli Bai Study. We are indebted to Claudia Olecjnizak, Richard Parnell, and Emma Stokes for ensuring collection of the long-term Mbeli Bai Study dataset, and to the research and support staff that enabled data collection, most notably Kelly Greenway, Jana Robeyst, and Vidrige Kandza. We thank Terry Brncic and Claudia Stephan for facilitating the collaboration, and Wildlife Conservation Society, Congo for logistical and administrative support. We thank all researchers involved in the collection of the Lokoué dataset, and are grateful to Magdalena Bermejo and Jake Dunn for their discussion and advice in the development of this research.

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
