## [Reviewer comments · Proceedings of the Royal Society B: Biological Sciences]

Review History

RSPB-2019-0681.R0 (Original submission)

Review form: Reviewer 1

Recommendation

Major revision is needed (please make suggestions in comments)

Scientific importance: Is the manuscript an original and important contribution to its field?

Acceptable

General interest: Is the paper of sufficient general interest?

Good

Quality of the paper: Is the overall quality of the paper suitable?

Acceptable

Is the length of the paper justified?

Yes

Should the paper be seen by a specialist statistical reviewer?

Yes

Do you have any concerns about statistical analyses in this paper? If so, please specify them explicitly in your report.

Yes

It is a condition of publication that authors make their supporting data, code and materials available - either as supplementary material or hosted in an external repository. Please rate, if applicable, the supporting data on the following criteria.

Is it accessible?

Yes

Is it clear?

Yes

Is it adequate?

Yes

Do you have any ethical concerns with this paper?

No

Comments to the Author

This manuscript empirically tests the idea that western gorillas, like humans, some other primates (e.g. hamadryas baboons, geladas) elephants, and whales, live in a hierarchically nested, sometimes-kin-structured social organization. It addresses a very interesting question about these understudied animals, and makes good use of the extensive bai data from multiple field sites.

The introduction to the paper is extremely well written. It is clear and concise and is a pleasure to read. I have only two comments/questions about the introduction. First, I am unclear on what the justification is for claiming that HSM is 'uniquely human' if we already know that it occurs in other primates, whales, and elephants. Is this claim perhaps outdated, written before we knew that it occurred in these other taxa? Please clarify. The answer to this is important, as I think it fundamentally affects whether this paper is best suited for Proceedings B, or some other journal. Second, I would suggest adding another sentence or two on western gorilla dispersal/social structure. For example, the term 'solitaries' comes up in the methods, but the authors did not clarify earlier that many adult males live alone. It doesn't need to be much, but for the very general readership of Proceedings B, it's best not to make any assumptions about what people do or do not know about gorillas.

Beyond the introduction, the paper needs extensive editing to make the methods and results intelligible to readers in a broad range of specialties. Relatively few Proceedings B readers will have much, if any, familiarity with the modeling framework the authors have adopted, and it is not currently explained in a manner that makes it accessible to anyone who is not already intimately familiar with the analysis methods. In general, the authors need to find a way to explain the statistical modeling in a way that tells us about the biological significance. E.g., what are bifurcation heights telling us? What about bifurcation differences? Are the SR and BP association indices simply complimentary ways of analyzing the data, or is there some specific purpose to using both? What are changes in gradients/knots telling us? Those are just a few examples. The way the statistical methods – and to some degree the results – are written, they are inaccessible to anyone who does not do exactly this sort of work themselves.

I think at least part of this could be remedied by including more basic description. While I sympathize with the authors' desire to be concise, what this functionally means is that I still don't have a clear idea of what the authors are proposing this hierarchical structure looks like. How big are these second-tier associations that the authors are newly describing? How closely related, on average, are the males who make up the members of a group at this tier? What might tie together the males who are members of a group at this tier for the Mbeli population, if not genetic kinship? It doesn't help much to say that these are equivalent to e.g. elephant clans or baboon gangs if we don't know what those are (and again, most people won't). The figures help only a little, if at all, since they seem to be missing captions, and a lot of the terms used in them are never really defined. As written, the biology has gotten lost underneath the numbers.

I would be happy to read a revised version of this article and offer more specific comments once I have a clearer idea of what was done and what was found. I could not agree with the authors more that an unquestioned assumption that chimpanzees are always the best model for understanding human evolution, based only on their genetic relatedness, is extremely problematic. We need more studies like these to clarify where humans do, and do not, overlap with the many interesting characteristics of great ape social behavior.

Review form: Reviewer 2 (Noah Snyder-Mackler)

Recommendation

Accept with minor revision (please list in comments)

Scientific importance: Is the manuscript an original and important contribution to its field?

Good

General interest: Is the paper of sufficient general interest?

Good

Quality of the paper: Is the overall quality of the paper suitable?

Good

Is the length of the paper justified?

Yes

Should the paper be seen by a specialist statistical reviewer?

Yes

Do you have any concerns about statistical analyses in this paper? If so, please specify them explicitly in your report.

Yes

It is a condition of publication that authors make their supporting data, code and materials available - either as supplementary material or hosted in an external repository. Please rate, if applicable, the supporting data on the following criteria.

Is it accessible?

Yes

Is it clear?

Yes

Is it adequate?

Yes

Do you have any ethical concerns with this paper?

No

Comments to the Author

In this manuscript the authors take a network-based approach to classify the hierarchical structure of western lowland gorillas. Overall, the manuscript is well written and the analyses relatively clearly presented. The authors also take care to place their work in the broader literature, and fill an important gap in our knowledge of the evolution of hierarchical social modularity (HSM) in apes (and therefore humans). Given the sophisticated approach and thorough background/context, this should be of broad interest to the readers of Proceedings B. I only have a few comments for the authors that they should address in any resubmission:

1. Overall, the description of the network methods are short and filled with jargon. It would be useful for the authors to clearly state what each approach brings to the table and, without using jargon, describe what each method does. For instance, the different clustering approaches in lines 100-110 could be explained more clearly. What does the "cluster_resolution" algorithm do? What do "multiple peaks in modularity" mean for the social structure of gorillas? Some, but not all of this detail, is present in the supplementary methods.
2. In the main text, the authors should clarify what each node in their network represents. In the main text it is unclear if each node is an individual silverback (or solitary male), or if each node in the network is a "group" or "solitary". I had to go to the supplementary tables to figure out that it was "group" & "solitary". For instance, in lines 130-132 they mention "silverbacks" and not "groups". Which is it? Please be consistent.
3. Lines 121-124: There is too little detail in the main methods here about the kinship coefficients and models. These details are in the SI, but should be moved here. For instance, your binarization of "related" ($r > 0.2$) vs. "unrelated" ($r < 0.2$), should be clearly explained in the main text. The relatedness analysis/results in Lines 193-195 are unclear. What is the predictor and what is the outcome variable? What is the "membership in the higher tier modules" and what about lower tiers?
4. Lines 149-152 (and elsewhere). I don't think that Kendall's rank correlation is an appropriate statistic here since the two vectors of association indices are not independent. A better approach would be to use a Mantel's test, which accounts for the interdependence of association indices in a matrix.
5. It would be useful to include comparisons to other primates with modular societies, like golden snub-nosed monkeys, geladas (the g3 level is similar to gelada "teams"), and uakaris.
6. This is probably due to the manuscript submission system, but there were no figure captions in the manuscript or online. This made it difficult to interpret the figures.

Decision letter (RSPB-2019-0681.R0)

26-Apr-2019

Dear Ms Morrison:

Your manuscript has now been peer reviewed and the reviews have been assessed by an Associate Editor. The reviewers' comments (not including confidential comments to the Editor) and the comments from the Associate Editor are included at the end of this email for your reference. As you will see, the reviewers and the Editors have raised some concerns with your manuscript and we would like to invite you to revise your manuscript to address them. In particular, please focus on clarifying your statistical approach, as outlined by both reviewers, and how your paper fits in to the broader literature on hierarchical social modularity across animal species.

Research ethics:

Use of animals and field studies:

If you wish to submit your data to Dryad (<http://datadryad.org/>) and have not already done so you can submit your data via this link [http://datadryad.org/submit?journalID=RSPB&manu=\(Document not available\)](http://datadryad.org/submit?journalID=RSPB&manu=(Document%20not%20available)), which will take you to your unique entry in the Dryad repository.

Please submit a copy of your revised paper within three weeks. If we do not hear from you within this time your manuscript will be rejected. If you are unable to meet this deadline please let us know as soon as possible, as we may be able to grant a short extension.

Best wishes,

Prof Sarah F Brosnan
Editor, Proceedings B
mailto: proceedingsb@royalsociety.org

Associate Editor
Comments to Author:

I have been fortunate to receive reviews of your article from two experts in the field. Both reviewers reported that your study was strong and should be of interest to a broad audience - ideal for the readership of this journal. I agree. However, both reviewers also requested that you

provide greater clarity in your description of your analytical approach and your interpretation of them. I agree that this will enhance your article and make it more accessible to a wide readership and that doing so will make it a stronger contribution to this journal.

Reviewer(s)' Comments to Author:

Referee: 1

Comments to the Author(s)

This manuscript empirically tests the idea that western gorillas, like humans, some other primates (e.g. hamadryas baboons, geladas) elephants, and whales, live in a hierarchically nested, sometimes-kin-structured social organization. It addresses a very interesting question about these understudied animals, and makes good use of the extensive bai data from multiple field sites.

The introduction to the paper is extremely well written. It is clear and concise and is a pleasure to read. I have only two comments/questions about the introduction. First, I am unclear on what the justification is for claiming that HSM is 'uniquely human' if we already know that it occurs in other primates, whales, and elephants. Is this claim perhaps outdated, written before we knew that it occurred in these other taxa? Please clarify. The answer to this is important, as I think it fundamentally affects whether this paper is best suited for Proceedings B, or some other journal. Second, I would suggest adding another sentence or two on western gorilla dispersal/social structure. For example, the term 'solitaries' comes up in the methods, but the authors did not clarify earlier that many adult males live alone. It doesn't need to be much, but for the very general readership of Proceedings B, it's best not to make any assumptions about what people do or do not know about gorillas.

Beyond the introduction, the paper needs extensive editing to make the methods and results intelligible to readers in a broad range of specialties. Relatively few Proceedings B readers will have much, if any, familiarity with the modeling framework the authors have adopted, and it is not currently explained in a manner that makes it accessible to anyone who is not already intimately familiar with the analysis methods. In general, the authors need to find a way to explain the statistical modeling in a way that tells us about the biological significance. E.g., what are bifurcation heights telling us? What about bifurcation differences? Are the SR and BP association indices simply complimentary ways of analyzing the data, or is there some specific purpose to using both? What are changes in gradients/knots telling us? Those are just a few examples. The way the statistical methods – and to some degree the results – are written, they are inaccessible to anyone who does not do exactly this sort of work themselves.

I think at least part of this could be remedied by including more basic description. While I sympathize with the authors' desire to be concise, what this functionally means is that I still don't have a clear idea of what the authors are proposing this hierarchical structure looks like. How big are these second-tier associations that the authors are newly describing? How closely related, on average, are the males who make up the members of a group at this tier? What might tie together the males who are members of a group at this tier for the Mbeli population, if not genetic kinship? It doesn't help much to say that these are equivalent to e.g. elephant clans or baboon gangs if we don't know what those are (and again, most people won't). The figures help only a little, if at all, since they seem to be missing captions, and a lot of the terms used in them are never really defined. As written, the biology has gotten lost underneath the numbers.

I would be happy to read a revised version of this article and offer more specific comments once I have a clearer idea of what was done and what was found. I could not agree with the authors

more that an unquestioned assumption that chimpanzees are always the best model for understanding human evolution, based only on their genetic relatedness, is extremely problematic. We need more studies like these to clarify where humans do, and do not, overlap with the many interesting characteristics of great ape social behavior.

Referee: 2

Comments to the Author(s)

In this manuscript the authors take a network-based approach to classify the hierarchical structure of western lowland gorillas. Overall, the manuscript is well written and the analyses relatively clearly presented. The authors also take care to place their work in the broader literature, and fill an important gap in our knowledge of the evolution of hierarchical social modularity (HSM) in apes (and therefore humans). Given the sophisticated approach and thorough background/context, this should be of broad interest to the readers of Proceedings B. I only have a few comments for the authors that they should address in any resubmission:

1. Overall, the description of the network methods are short and filled with jargon. It would be useful for the authors to clearly state what each approach brings to the table and, without using jargon, describe what each method does. For instance, the different clustering approaches in lines 100-110 could be explained more clearly. What does the “cluster_resolution” algorithm do? What do “multiple peaks in modularity” mean for the social structure of gorillas? Some, but not all of this detail, is present in the supplementary methods.
2. In the main text, the authors should clarify what each node in their network represents. In the main text it is unclear if each node is an individual silverback (or solitary male), or if each node in the network is a “group” or “solitary”. I had to go to the supplementary tables to figure out that it was “group” & “solitary”. For instance, in lines 130-132 they mention “silverbacks” and not “groups”. Which is it? Please be consistent.
3. Lines 121-124: There is too little detail in the main methods here about the kinship coefficients and models. These details are in the SI, but should be moved here. For instance, your binarization of “related” ($r > 0.2$) vs. “unrelated” ($r < 0.2$), should be clearly explained in the main text. The relatedness analysis/results in Lines 193-195 are unclear. What is the predictor and what is the outcome variable? What is the “membership in the higher tier modules” and what about lower tiers?
4. Lines 149-152 (and elsewhere). I don’t think that Kendall’s rank correlation is an appropriate statistic here since the two vectors of association indices are not independent. A better approach would be to use a Mantel’s test, which accounts for the interdependence of association indices in a matrix.
5. It would be useful to include comparisons to other primates with modular societies, like golden snub-nosed monkeys, geladas (the g3 level is similar to gelada “teams”), and uakaris.
6. This is probably due to the manuscript submission system, but there were no figure captions in the manuscript or online. This made it difficult to interpret the figures.

Author's Response to Decision Letter for (RSPB-2019-0681.R0)

See Appendix A.

RSPB-2019-0681.R1 (Revision)

Review form: Reviewer 1

Recommendation

Accept with minor revision (please list in comments)

Scientific importance: Is the manuscript an original and important contribution to its field?

Good

General interest: Is the paper of sufficient general interest?

Excellent

Quality of the paper: Is the overall quality of the paper suitable?

Excellent

Is the length of the paper justified?

Yes

Should the paper be seen by a specialist statistical reviewer?

Yes

Do you have any concerns about statistical analyses in this paper? If so, please specify them explicitly in your report.

Yes

It is a condition of publication that authors make their supporting data, code and materials available - either as supplementary material or hosted in an external repository. Please rate, if applicable, the supporting data on the following criteria.

Is it accessible?

Yes

Is it clear?

Yes

Is it adequate?

Yes

Do you have any ethical concerns with this paper?

No

Comments to the Author

The authors have done a reasonably satisfactory job of addressing both my concerns, and the concerns of reviewer #2. Their clarifications do help, though I think it would be useful to move some of the information in the supplementary materials to the main text – I am thinking especially of Figure S2 and its accompanying information, which are an important part of the reason this paper is of broad interest.

If the authors are given another chance to revise, then I would also encourage them to work on clarifying their figure captions. These are still lacking much, if any, biology, and largely do not

address the all-important question of what the figure means. Thanks to the clarifications in the text it is now easier to figure out, but a one-sentence description of the biology in each of their figure captions would go a long way. More specific axis titles might also help—e.g. distance from what? Probability of what? It should not be necessary to hunt through the text to figure this out.

As I mentioned in my comments to the editor during the first round of reviews, I believe that this paper should also be evaluated by someone with greater expertise in the statistical techniques the authors are using. If reviewer #2 has such expertise, wonderful, but I myself do not feel qualified to say with any authority that the clustering and network modularity analyses were handled appropriately.

That said, this is an interesting paper with important ramifications for our understanding of a phenomenon—HSM—that has received much attention in the cognition and evolution literatures. I thus feel that Proceedings B is an appropriate home for it, and would recommend acceptance.

Review form: Reviewer 2 (Noah Snyder-Mackler)

Recommendation

Accept as is

Scientific importance: Is the manuscript an original and important contribution to its field?

Good

General interest: Is the paper of sufficient general interest?

Acceptable

Quality of the paper: Is the overall quality of the paper suitable?

Good

Is the length of the paper justified?

Yes

Should the paper be seen by a specialist statistical reviewer?

No

Do you have any concerns about statistical analyses in this paper? If so, please specify them explicitly in your report.

No

It is a condition of publication that authors make their supporting data, code and materials available - either as supplementary material or hosted in an external repository. Please rate, if applicable, the supporting data on the following criteria.

Is it accessible?

Yes

Is it clear?

Yes

Is it adequate?

No

Do you have any ethical concerns with this paper?

No

Comments to the Author

The authors have thoughtfully revised the manuscript and I have no further comments.

Decision letter (RSPB-2019-0681.R1)

14-Jun-2019

Dear Ms Morrison

I am pleased to inform you that your Review manuscript RSPB-2019-0681.R1 entitled "Hierarchical Social Modularity in Gorillas" has been accepted for publication in Proceedings B, pending some minor revisions. In particular, please consider moving the supplemental information Reviewer 1 highlights to the main body of your paper, clarify your Figure captions as per the reviewer's comments, and make your R code accessible. You can find the additional specific requests in the reviewers' and AE's comments below. Because the schedule for publication is very tight, it is a condition of publication that you submit the revised version of your manuscript within 7 days. If you do not think you will be able to meet this date please let me know immediately.

To upload your manuscript, log into <http://mc.manuscriptcentral.com/prsb> and enter your Author Centre, where you will find your manuscript title listed under "Manuscripts with Decisions." Under "Actions," click on "Create a Revision." Your manuscript number has been appended to denote a revision.

You will be unable to make your revisions on the originally submitted version of the manuscript. Instead, upload a new version through your Author Centre.

- 1) A text file of the manuscript (doc, txt, rtf or tex), including the references, tables (including captions) and figure captions. Please remove any tracked changes from the text before submission. PDF files are not an accepted format for the "Main Document".
- 2) A separate electronic file of each figure (tiff, EPS or print-quality PDF preferred). The format should be produced directly from original creation package, or original software format. Please note that PowerPoint files are not accepted.
- 3) Electronic supplementary material: this should be contained in a separate file from the main text and the file name should contain the author's name and journal name, e.g `authorname_procb_ESM_figures.pdf`

All supplementary materials accompanying an accepted article will be treated as in their final form. They will be published alongside the paper on the journal website and posted on the online figshare repository. Files on figshare will be made available approximately one week before the accompanying article so that the supplementary material can be attributed a unique DOI. Please see: <https://royalsociety.org/journals/authors/author-guidelines/>

4) Data-Sharing and data citation

It is a condition of publication that data supporting your paper are made available. Data should be made available either in the electronic supplementary material or through an appropriate repository. Details of how to access data should be included in your paper. Please see <https://royalsociety.org/journals/ethics-policies/data-sharing-mining/> for more details.

If you wish to submit your data to Dryad (<http://datadryad.org/>) and have not already done so you can submit your data via this link <http://datadryad.org/submit?journalID=RSPB&manu=RSPB-2019-0681.R1> which will take you to your unique entry in the Dryad repository.

Once again, thank you for submitting your manuscript to Proceedings B and I look forward to receiving your final version. If you have any questions at all, please do not hesitate to get in touch.

Sincerely,

Dr Sarah Brosnan
Editor, Proceedings B
<mailto:proceedingsb@royalsociety.org>

Associate Editor Board Member: 1

Comments to Author:

Thank you very much for making substantial changes to your article. The two reviewers who reviewed your original submission kindly reviewed this revision. Both agreed that your article was much improved and recommended it for publication. I concur. To finalize your article for acceptance, can you please enhance the clarity of your figure captions as noted by one of the reviewers. Additionally, for full clarity, can you please also provide your (analysis) code, either as supplementary materials or shared via a repository such as GitHub. With those additions, it would be my pleasure to recommend your article for publication in Proc R Soc B.

Reviewer(s)' Comments to Author:

Referee: 2

Comments to the Author(s)

The authors have thoughtfully revised the manuscript and I have no further comments.

Referee: 1

Comments to the Author(s)

The authors have done a reasonably satisfactory job of addressing both my concerns, and the concerns of reviewer #2. Their clarifications do help, though I think it would be useful to move

some of the information in the supplementary materials to the main text – I am thinking especially of Figure S2 and its accompanying information, which are an important part of the reason this paper is of broad interest.

If the authors are given another chance to revise, then I would also encourage them to work on clarifying their figure captions. These are still lacking much, if any, biology, and largely do not address the all-important question of what the figure means. Thanks to the clarifications in the text it is now easier to figure out, but a one-sentence description of the biology in each of their figure captions would go a long way. More specific axis titles might also help – e.g. distance from what? Probability of what? It should not be necessary to hunt through the text to figure this out.

As I mentioned in my comments to the editor during the first round of reviews, I believe that this paper should also be evaluated by someone with greater expertise in the statistical techniques the authors are using. If reviewer #2 has such expertise, wonderful, but I myself do not feel qualified to say with any authority that the clustering and network modularity analyses were handled appropriately.

That said, this is an interesting paper with important ramifications for our understanding of a phenomenon – HSM – that has received much attention in the cognition and evolution literatures. I thus feel that Proceedings B is an appropriate home for it, and would recommend acceptance.

Decision letter (RSPB-2019-0681.R2)

20-Jun-2019

Dear Ms Morrison

I am pleased to inform you that your manuscript entitled "Hierarchical Social Modularity in Gorillas" has been accepted for publication in Proceedings B.

Your article has been estimated as being 9 pages long. Our Production Office will be able to confirm the exact length at proof stage.

Open Access

Paper charges

Sincerely,

Editor, Proceedings B
mailto: proceedingsb@royalsociety.org

Appendix A

Associate Editor

Comments to Author:

I have been fortunate to receive reviews of your article from two experts in the field. Both reviewers reported that your study was strong and should be of interest to a broad audience - ideal for the readership of this journal. I agree. However, both reviewers also requested that you provide greater clarity in your description of your analytical approach and your interpretation of them. I agree that this will enhance your article and make it more accessible to a wide readership and that doing so will make it a stronger contribution to this journal.

Extensive additions to the methods section to make them more accessible, and clarification within the results section

Referee: 1

Comments to the Author(s)

This manuscript empirically tests the idea that western gorillas, like humans, some other primates (e.g. hamadryas baboons, geladas) elephants, and whales, live in a hierarchically nested, sometimes-kin-structured social organization. It addresses a very interesting question about these understudied animals, and makes good use of the extensive bai data from multiple field sites.

The introduction to the paper is extremely well written. It is clear and concise and is a pleasure to read. I have only two comments/questions about the introduction. First, I am unclear on what the justification is for claiming that HSM is 'uniquely human' if we already know that it occurs in other primates, whales, and elephants. Is this claim perhaps outdated, written before we knew that it occurred in these other taxa? Please clarify. The answer to this is important, as I think it fundamentally affects whether this paper is best suited for Proceedings B, or some other journal.

Hierarchical social modularity is not unique to humans, but rather what has been suggested in the past is that the form of hierarchical social modularity observed in humans is qualitatively different to that in other species. Humans show a greater number of levels of social structure and cooperation between large numbers of unrelated individuals. This was thought to be evolutionarily distinct from HSM in other species, as no other apes showed the behaviour, suggesting it evolved separately and was thought to have been enabled by brain expansion during hominin evolution. In the sentence: "An extreme version of the social brain narrative is that human HSM is a unique product of hominin brain evolution" We mean to communicate how human HSM was thought to be unique from HSM in other species, but can see how that can easily be misinterpreted, so have removed the word unique and added "distinct from HSM observed in other primates" (line 41-42)

Second, I would suggest adding another sentence or two on western gorilla dispersal/social structure. For example, the term 'solitaries' comes up in the methods, but the authors did not clarify earlier that many adult males live alone. It doesn't need to be much, but for the very general readership of Proceedings B, it's best not to make any assumptions about what people do or do not know about gorillas.

Extra information on gorilla social structure and dispersion added in lines 55-58

Beyond the introduction, the paper needs extensive editing to make the methods and results intelligible to readers in a broad range of specialties. Relatively few Proceedings B readers

will have much, if any, familiarity with the modeling framework the authors have adopted, and it is not currently explained in a manner that makes it accessible to anyone who is not already intimately familiar with the analysis methods. In general, the authors need to find a way to explain the statistical modeling in a way that tells us about the biological significance. E.g., what are bifurcation heights telling us? What about bifurcation differences? Are the SR and BP association indices simply complimentary ways of analyzing the data, or is there some specific purpose to using both? What are changes in gradients/knots telling us? Those are just a few examples. The way the statistical methods—and to some degree the results—are written, they are inaccessible to anyone who does not do exactly this sort of work themselves. *Extensive additions to the methods section to make them more accessible*

I think at least part of this could be remedied by including more basic description. While I sympathize with the authors' desire to be concise, what this functionally means is that I still don't have a clear idea of what the authors are proposing this hierarchical structure looks like. How big are these second-tier associations that the authors are newly describing? *Inclusion of the number of gorilla groups, solitaries and individual gorillas in each level of social structure detected (lines 178, 182-183 and 208)*

How closely related, on average, are the males who make up the members of a group at this tier? What might tie together the males who are members of a group at this tier for the Mbeli population, if not genetic kinship? It doesn't help much to say that these are equivalent to e.g. elephant clans or baboon gangs if we don't know what those are (and again, most people won't). *Additional information lines 245-255*

The figures help only a little, if at all, since they seem to be missing captions, and a lot of the terms used in them are never really defined. As written, the biology has gotten lost underneath the numbers. *Captions now listed at end of manuscript*

I would be happy to read a revised version of this article and offer more specific comments once I have a clearer idea of what was done and what was found. I could not agree with the authors more that an unquestioned assumption that chimpanzees are always the best model for understanding human evolution, based only on their genetic relatedness, is extremely problematic. We need more studies like these to clarify where humans do, and do not, overlap with the many interesting characteristics of great ape social behavior.

Referee: 2

Comments to the Author(s)

In this manuscript the authors take a network-based approach to classify the hierarchical structure of western lowland gorillas. Overall, the manuscript is well written and the analyses relatively clearly presented. The authors also take care to place their work in the broader literature, and fill an important gap in our knowledge of the evolution of hierarchical social modularity (HSM) in apes (and therefore humans). Given the sophisticated approach and thorough background/context, this should be of broad interest to the readers of Proceedings B. I only have a few comments for the authors that they should address in any resubmission:

1. Overall, the description of the network methods are short and filled with jargon. It would be

useful for the authors to clearly state what each approach brings to the table and, without using jargon, describe what each method does. For instance, the different clustering approaches in lines 100-110 could be explained more clearly. What does the “cluster_resolution” algorithm do? What do “multiple peaks in modularity” mean for the social structure of gorillas? Some, but not all of this detail, is present in the supplementary methods. *Extensive additions to the methods section to make them more accessible.*

2. In the main text, the authors should clarify what each node in their network represents. In the main text it is unclear if each node is an individual silverback (or solitary male), or if each node in the network is a “group” or “solitary”. I had to go to the supplementary tables to figure out that it was “group” & “solitary”. For instance, in lines 130-132 they mention “silverbacks” and not “groups”. Which is it? Please be consistent. *We have now changed every use of “silverback” to “group or solitary male”. We had used silverback to be more concise when discussing silverback male gorillas and their associated groups as well as solitary silverback males but can see that this is likely to cause unnecessary confusion, especially in those without considerable background knowledge on gorilla social structure.*

3. Lines 121-124: There is too little detail in the main methods here about the kinship coefficients and models. These details are in the SI, but should be moved here. For instance, your binarization of “related” ($r > 0.2$) vs. “unrelated” ($r < 0.2$), should be clearly explained in the main text. The relatedness analysis/results in Lines 193-195 are unclear. What is the predictor and what is the outcome variable? What is the “membership in the higher tier modules” and what about lower tiers?

*Lines 158-163 to provide more information on kinship methods.
Extra clarification added to results in lines 236-241*

4. Lines 149-152 (and elsewhere). I don't think that Kendall's rank correlation is an appropriate statistic here since the two vectors of association indices are not independent. A better approach would be to use a Mantel's test, which accounts for the interdependence of association indices in a matrix.

Yes a Mantel's test looks like a better statistic to use. We have therefore reanalysed the data using a mantels test to demonstrate consistency between time periods (lines 191-194) and consistency between same day presence vs presence within 100m (supplementary information)

5. It would be useful to include comparisons to other primates with modular societies, like golden snub-nosed monkeys, geladas (the g3 level is similar to gelada "teams"), and uakaris. *Baboons and now gelada social levels are directly compared with those detected in gorillas here (lines 217-219). There are such a large number of primate species showing modular societies that we don't want to list them all but have instead added an extra sentence guiding the reader to a paper that lists all primate modular societies (line 261-262).*

6. This is probably due to the manuscript submission system, but there were no figure captions in the manuscript or online. This made it difficult to interpret the figures. *Captions now listed at end of manuscript*